# GhostNetV2: Enhance Cheap Operation with Long-Range Attention

**Yehui Tang**[1,2], **Kai Han**[2], **Jianyuan Guo**[2,3], **Chang Xu**[3], **Chao Xu**[1], **Yunhe Wang**[2*]

[1]School of Artificial Intelligence, Peking University [2]Huawei Noah's Ark Lab
[3]School of Computer Science, University of Sydney
yhtang@pku.edu.cn, {kai.han, yunhe.wang}@huawei.com

## Abstract

Light-weight convolutional neural networks (CNNs) are specially designed for applications on mobile devices with faster inference speed. The convolutional operation can only capture local information in a window region, which prevents performance from being further improved. Introducing self-attention into convolution can capture global information well, but it will largely encumber the actual speed. In this paper, we propose a hardware-friendly attention mechanism (dubbed DFC attention) and then present a new GhostNetV2 architecture for mobile applications. The proposed DFC attention is constructed based on fully-connected layers, which can not only execute fast on common hardware but also capture the dependence between long-range pixels. We further revisit the expressiveness bottleneck in previous GhostNet and propose to enhance expanded features produced by cheap operations with DFC attention, so that a GhostNetV2 block can aggregate local and long-range information simultaneously. Extensive experiments demonstrate the superiority of GhostNetV2 over existing architectures. For example, it achieves 75.3% top-1 accuracy on ImageNet with 167M FLOPs, significantly suppressing Ghost-NetV1 (74.5%) with a similar computational cost. The source code will be available at `https://github.com/huawei-noah/Efficient-AI-Backbones/tree/master/ghostnetv2_pytorch` and `https://gitee.com/mindspore/models/tree/master/research/cv/ghostnetv2`.

## 1 Introduction

In computer vision, the architecture of deep neural network plays a vital role for various tasks, such as image classification [19, 10], object detection [27, 26], and video analysis [18]. In the past decade, the network architecture has been evolving rapidly, and a series of milestones including AlexNet [19], GoogleNet [29], ResNet [10] and EfficientNet [32] have been developed. These networks have pushed the performances of a wide range of visual tasks to a high level.

To deploy neural networks on edge devices like smartphone and wearable devices, we need to consider not only the performance of a model, but also its efficiency especially the actual inference speed. Matrix multiplications occupy the main part of computational cost and parameters. Developing light-weight models is a promising approach to reduce the inference latency. MobileNet [13] factorizes a standard convolution into depthwise convolution and point-wise convolution, which reduces the computational cost drastically. MobileNetV2 [28] and MobileNetV3 [12] further introduce the inverted residual block and improve the network architecture. ShuffleNet [42] utilizes the shuffle operation to encourage the information exchange between channel groups. GhostNet [8] proposes the cheap operation to reduce feature redundancy in channels. WaveMLP [33] replaces the complex

---

*Corresponding author.

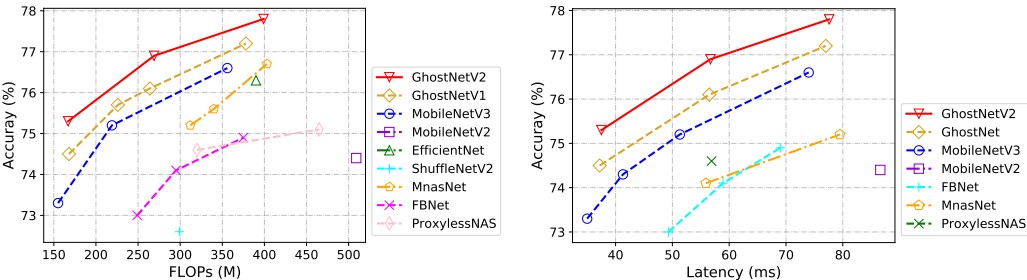

Figure 1: Top-1 accuracy *vs.*FLOPs on ImageNet dataset.

Figure 2: Top-1 accuracy *vs.*latency on ImageNet dataset.

self-attention module with a simple Multi-Layer Perceptron (MLP) to reduce the computational cost. These light-weight neural networks have been applied in many mobile applications.

Nevertheless, the convolution-based light-weight models are weak in modeling long-range dependency, which limits further performance improvement. Recently, transformer-like models are introduced to computer vision, in which the self-attention module can capture the global information. The typical self-attention module requires quadratic complexity *w.r.t.* the size of feature's shape and is not computationally friendly. Moreover, plenty of feature splitting and reshaping operations are required to calculate the attention map. Though their theoretical complexity is negligible, these operations incur more memory usage and longer latency in practice. Thus, utilizing vanilla self-attention in light-weight models is not friendly for mobile deployment. For example, MobileViT with massive self-attention operations is more than $7\times$ slower than MobileNetV2 on ARM devices [23].

In this paper, we propose a new attention mechanism (dubbed DFC attention) to capture the long-range spatial information, while keeping the implementation efficiency of light-weight convolutional neural networks. Only fully connected (FC) layers participate in generating the attention maps for simplicity. Specifically, a FC layer is decomposed into horizontal FC and vertical FC to aggregate pixels in a 2D feature map of CNN. The two FC layers involve pixels in a long range along their respective directions, and stacking them will produce a global receptive field. Moreover, starting from ate-of-the-art GhostNet, we revisit its representation bottleneck and enhance the intermediate features with the DFC attention. Then we construct a new light-weight vision backbone, GhostNetV2. Compared with the existing architectures, it can achieve a better tread-off between accuracy and inference speed (as shown in Figures 1 and 2).

## 2   Related Work

It is a challenge to design a light-weight neural architecture with fast inference speed and high performance simultaneously [16, 41, 13, 40, 35]. SqueezeNet [16] proposes three strategies to design a compact model, *i.e.*, replacing $3 \times 3$ filters with $1 \times 1$ filers, decreasing the number of input channels to $3x3$ filters, and down-sampling late in the network to keep large feature maps. These principles are constructive, especially the usage of $1 \times 1$ convolution. MobileNetV1 [13] replaces almost all the $3 \times 3$ filers with $1 \times 1$ kernel and depth-wise separable convolutions, which dramatically reduces the computational cost. MobileNetV2 [28] further introduces the residual connection to the light-weight model, and constructs an inverted residual structure, where the intermediate layer of a block has more channels than its input and output. To keep representation ability, a part of non-linear functions are removed. MobileNeXt [44] rethinks the necessary of inverted bottleneck, and claims that the classic bottleneck structure can also achieve high performance. Considering the $1 \times 1$ convolution account for a substantial part of computational cost, ShuffleNet [42] replace it with group convolution. The channel shuffle operation to help the information flowing across different groups. By investigating the factors that affect the practical running speed, ShuffleNet V2 [22] proposes a hardware-friendly new block. By leveraging the feature's redundancy, GhostNet [8] replaces half channels in $1 \times 1$ convolution with cheap operations. Until now, GhostNet has been the SOTA light-weight model with a good trade-off between accuracy and speed.

Besides manual design, a series of methods try to search for a light-weight architecture. For example, FBNet [39] designs a hardware-aware searching strategy, which can directly find a good trade-off between accuracy and speed on a specific hardware. Based on the inverted residual bottleneck, MnasNet [31], MobileNetV3 [12] search the architecture parameters,such as model width, model depth, convolutional filter's size, *etc*. Though NAS based methods achieve high performance, their success is based on well-designed search spaces and architectural units. Automatic searching and manual design can be combined to find a better architecture.

## 3  Preliminary

### 3.1  A Brief Review of GhostNet

GhostNet [8] is SOTA light-weight model designed for efficient inference on mobile devices. Its main component is the Ghost module, which can replace the original convolution by generating more feature maps from cheap operations. Given input feature $X \in \mathbb{R}^{H \times W \times C}$ with height $H$, width $W$ and channel's number $C$, a typical Ghost module can replace a standard convolution by two steps. Firstly, a $1 \times 1$ convolution is used to generate the intrinsic feature, *i.e.*,

$$Y' = X * F_{1 \times 1}, \tag{1}$$

where $*$ denotes the convolution operation. $F_{1 \times 1}$ is the point-wise convolution, and $Y' \in \mathbb{R}^{H \times W \times C'_{out}}$ is the intrinsic features, whose sizes are usually smaller than the original output features, *i.e.*, $C'_{out} < C_{out}$. Then cheap operations (*e.g.*, depth-wise convolution) are used to generate more features based on the intrinsic features. The two parts of features are concatenated along the channel dimension, *i.e.*,

$$Y = \text{Concat}([Y', Y' * F_{dp}]), \tag{2}$$

where $F_{dp}$ is the depth-wise convolutional filter, and $Y \in \mathbb{R}^{H \times W \times C_{out}}$ is the output feature. Though Ghost module can reduce the computational cost significantly, the representation ability is inevitably weakened. The relationship between spatial pixels is vital to make accurate recognition. While in GhostNet, the spatial information is only captured by the cheap operations (usually implemented by $3 \times 3$ depth-wise convolution) for half of the features. The remaining features are just produced by $1 \times 1$ point-wise convolution, without any interaction with other pixels. The weak ability to capture the spatial information may prevent performance from being further improved.

A block of GhostNet is constructed by stacking two Ghost modules (shown in Figure 4(a)). Similar to MobileNetV2 [28], it is also an inverted bottleneck, *i.e.*, the first Ghost module acts as an expansion layer to increase the number of output channels, and the second Ghost module reduces the channels' number to match the shortcut path.

### 3.2  Revisit Attention for Mobile Architecture

Originating from the NLP field [36], attention-based models are introduced to computer vision recently [6, 9, 34, 7]. For example, ViT [6] uses the standard transformer model stacked by self-attention modules and MLP modules. Wang *et al.*insert the self-attention operation into convolutional neural networks to capture the non-local information [37]. A typical at-

Table 1: The comparison of theoretical FLOPs and practical latency.

| Model | Top-1 Acc. (%) | FLOPs (M) | Latency (ms) |
|---|---|---|---|
| GhostNet | 73.9 | 141 | 31.1 |
| + Self Attention [23] | 74.4 | 172 | 72.3 |
| + DFC Attention (Ours) | 75.3 | 167 | 37.5 |

tention module usually has a quadratic complexity *w.r.t.* the feature's size, which is unscalable to high-resolution images in downstream tasks such as object detection and semantic segmentation.

A mainstream strategy to reduce attention's complexity is splitting images into multiple windows and implementing the attention operation inside windows or crossing windows. For example, Swin Transformer [21] splits the original feature into multiple non-overlapped windows, and the self-attention is calculated within the local windows. MobileViT [23] also unfolds the feature into non-overlapping patches and calculates the attention across these patches. For the 2D feature map in CNN, implementing the feature splitting and attention calculation involves plenty of tensor reshaping

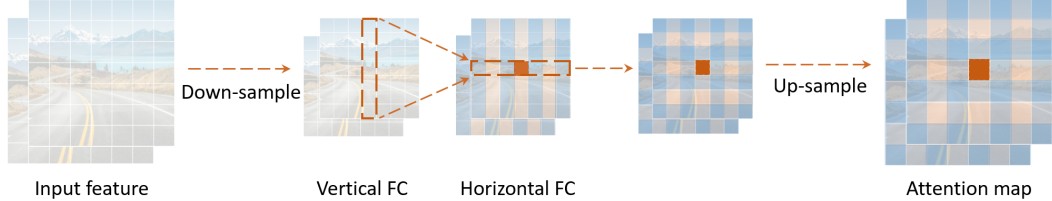

Down-sample             Up-sample

Input feature       Vertical FC     Horizontal FC            Attention map

Figure 3: The information flow of DFC attention. The horizontal and vertical FC layers capture the long-range information along the two directions, respectively.

and transposing operations. whose theoretical complexity is negligible. In a large model (*e.g.*, Swin-B [21] with several billion FLOPs) with high complexity, these operations only occupy a few portions of the total inference time. While for the light-weight models, their deploying latency cannot be overlooked.

For an intuitive understanding, we equip the GhostNet model with the self-attention used in MobileViT [23] and measure the latency on Huawei P30 (Kirin 980 CPU) with TFLite tool. We use the standard input's resolution of ImageNet, *i.e.*, $224 \times 224$, and show the results in Table 1. The attention mechanism only adds about 20% theoretical FLOPs, but requires $2\times$ inference time on a mobile device. The large difference between theoretical and practical complexity shows that it is necessary to design a hard-ware friendly attention mechanism for fast implementation on mobile devices.

## 4 Approach

### 4.1 DFC Attention for Mobile Architecture

In this section, we will discuss how to design an attention module for mobile CNNs. A desired attention is expected to have the following properties:

- **Long-range.** It is vital to capture the long-range spatial information for attention to enhance the representation ability, as a light-weight CNN (*e.g.*, MobileNet [13], GhostNet [8]) usually adopts small convolution filters (*e.g.*, $1 \times 1$ convolution) to save computational cost.

- **Deployment-efficient.** The attention module should be extremely efficient to avoid slowing the inference down. Expensive transformations with high FLOPs or hardware-unfriendly operations are unexpected.

- **Concept-simple.** To keep the model's generalization on diverse tasks, the attention module should be conceptually-simple with little dainty design.

Though self-attention operations [6, 24, 21] can model the long-range dependence well, they are not deployment-efficient as discussed in the above section. Compared with them, fully-connected (FC) layers with fixed weights are simpler and easier to implement, which can also be used to generate attention maps with global receptive fields. The detailed computational process is illustrated as follows.

Given a feature $Z \in \mathbb{R}^{H \times W \times C}$, it can be seen as $HW$ tokens $z_i \in \mathbb{R}^C$, *i.e.*, $Z = \{z_{11}, z_{12}, \cdots, z_{HW}\}$. A direct implementation of FC layer to generate the attention map is formulated as:

$$a_{hw} = \sum_{h',w'} F_{hw,h'w'} \odot z_{h'w'}, \tag{3}$$

where $\odot$ is element-wise multiplication, $F$ is the learnable weights in the FC layer, and $A = \{a_{11}, a_{12}, \cdots, a_{HW}\}$ is the generated attention map. Eq 3 can capture the global information by aggregating all the tokens together with learnable weights, which is much simpler than the typical self-attention [36] as well. However, its computational process still requires quadratic complexity *w.r.t.* feature's size (*i.e.*, $\mathcal{O}(H^2W^2))^2$, which is unacceptable in practical scenarios especially when the input images are of high resolutions. For example, the 4-th layer of GhostNet has a feature map

---

[2]The computational complexity *w.r.t.* channel's number $C$ is omitted for brevity.

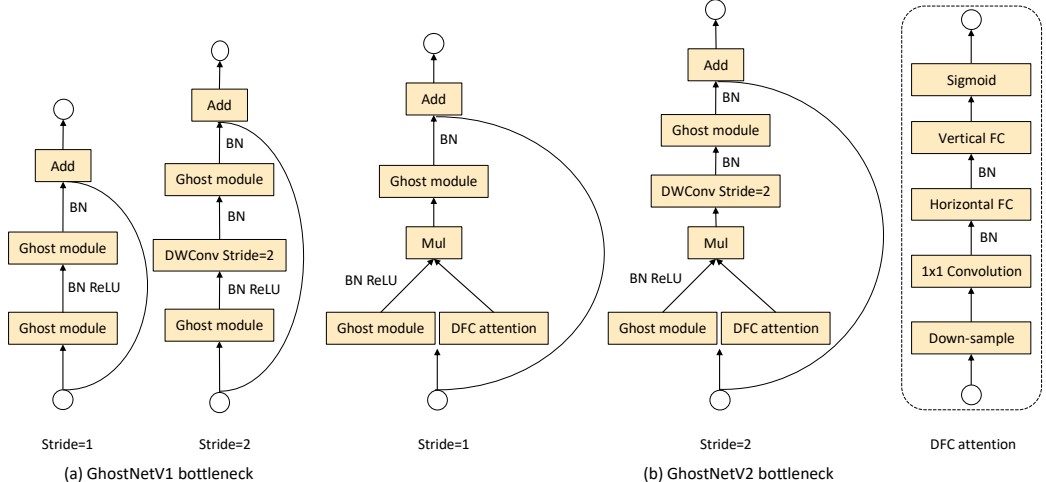

Figure 4: The diagrams of blocks in GhostNetV1 and GhostNetV2. Ghost block is an inverted residual bottleneck containing two Ghost modules, where DFC attention enhances the expanded features to improve expressiveness ability.

with 3136 ($56 \times 56$) tokens, which incurs prohibitively high complexity to calculate the attention map. Actually, feature maps in a CNN are usually of low-rank [30, 17], it is unnecessary to connect all the input and output tokens in different spatial locations densely. The feature's 2D shape naturally provides a perspective to reduce the computation of FC layers, *i.e.*, decomposing Eq. 3 into two FC layers and aggregating features along the horizontal and vertical directions, respectively. It can be formulated as:

$$a'_{hw} = \sum_{h'=1}^{H} F^H_{h,h'w} \odot z_{h'w}, h = 1, 2, \cdots, H, w = 1, 2, \cdots, W, \tag{4}$$

$$a_{hw} = \sum_{w'=1}^{W} F^W_{w,hw'} \odot a'_{hw'}, h = 1, 2, \cdots, H, w = 1, 2, \cdots, W, \tag{5}$$

where $F^H$ and $F^W$ are transformation weights. Taking the original feature $Z$ as input, Eq. 4 and Eq. 5 are applied to the features sequentially, capturing the long-range dependence along the two directions, respectively. We dub this operation as decoupled fully connected (DFC) attention, whose information flow is shown in Figure 3. Owing to the decoupling of horizontal and vertical transformations, the computational complexity of the attention module can be reduced to $\mathcal{O}(H^2W + HW^2)$. In the full attention (Eq. 3), all the patches in a square region participate in the calculation of the focused patch directly. In DFC attention, a patch is directly aggregated by patches in its vertical/horizontal lines, while other patches participate in the generation of those patches in the vertical/horizontal lines, having an indirect relationship with the focused token. Thus the calculation of a patch also involves all the patches in the square region.

Eqs. 4 and 5 denote the general formulation of DFC attention, which aggregates pixels along horizontal and vertical directions, respectively. By sharing a part of transformation weights, it can be conveniently implemented with convolutions, leaving out the time-consuming tensor reshaping and transposing operations that affect the practical inference speed. To process input images with varying resolutions, the filter's size can be decoupled with feature map's size, *i.e.*, two depth-wise convolutions with kernel sizes $1 \times K_H$ and $K_W \times 1$ are sequentially applied on the input feature. When implemented with convolution, the theoretical complexity of DFC attention is denoted as $\mathcal{O}(K_H HW + K_W HW)$. This strategy is well supported by tools such as TFLite and ONNX for fast inference on mobile devices.

## 4.2   GhosetNet V2

In this section, we use the DFC attention to improve the representation ability of lightweight models and then present the new vision backbone, GhostNetV2.

**Enhancing Ghost module.** As discussed in 3.1, only half of features in Ghost module (Eqs. 1 and 2) interact with other pixels, which damages its ability to capture spatial information. Hence we use DFC attention to enhance Ghost module's output feature $Y$ for capturing long-range dependence among different spatial pixels.

The input feature $X \in \mathbb{R}^{H \times W \times C}$ is sent to two branches, *i.e.*, one is the Ghost module to produce output feature $Y$ (Eqs. 1 and 2), and the other is the DFC module to generate attention map $A$ (Eqs. 4 and 5). Recalling that in a typical self-attention [36], linear transformation layers are used to transform input feature into query and key for calculating attention maps. Similarly, we also implement a $1 \times 1$ convolution to convert module's input $X$ into DFC's input $Z$. The final output $O \in \mathbb{R}^{H \times W \times C}$ of the module is the product of two branch's output, *i.e.*,

$$O = \text{Sigmoid}(A) \odot \mathcal{V}(X), \tag{6}$$

where $\odot$ is the element-wise multiplication and $\text{Sigmoid}$ is the scaling function to normalize the attention map $A$ into range $(0, 1)$.

The information aggregation process is shown in Figure 5. With the same input, the Ghost module and DFC attention are two parallel branches extracting information from different perspectives. The output is their element-wise product, which contains information from both features of the Ghost module and attentions of the DFC attention module. The calculation of each attention value involves patches in a large range so that the output feature can contain information from these patches.

**Feature downsampling.** As Ghost module (Eqs. 1 and 2) is an extremely efficient operation, directly paralleling the DFC attention with it will introduces extra computational cost. Hence we reduce the feature's size by down-sampling it both horizontally and vertically, so that all the operations in DFC attention can be conducted on the smaller features. By default, the width and height are both scaled to half of their original lengths, which reduces 75% FLOPs of DFC attention. Then produced feature map is then up-sampled to the original size to match the feature's size in Ghost branch. We naively use the average pooling and bilinear interpolation for downsampling and upsampling, respectively. Noticing that directly implementing sigmoid (or hard sigmoid) function will incur longer latency, we also deploy the sigmoid function on the downsampled features to accelerate practical inference. Though the value of attention maps may not be limited in range (0,1) strictly, we empirically find that its impact on the final performance is negligible.

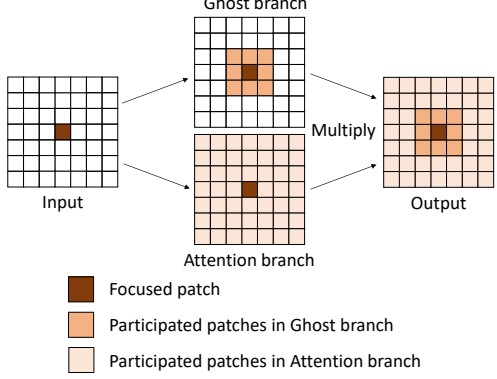

Figure 5: The information aggregation process of different patches.

**GhostV2 bottleneck.** GhostNet adopts an inverted residual bottleneck containing two Ghost modules, where the first module produces expanded features with more channels, while the second one reduces channel's number to get output features. This inverted bottleneck naturally decouples the "expressiveness" and "capacity" of a model [28]. The former is measured by the expanded features while the latter is reflected by the input/output domains of a block. The original Ghost module generates partial features via cheap operations, which damages both the expressiveness and the capacity. By investigating the performance difference of equipping DFC attention on the expanded features or output features (Table 8 in Section 5.4), we find that enhancing 'expressiveness' is more effective. Hence we only multiply the expanded features with DFC attention.

Figure 4(b) shows the diagram of GhostV2 bottleneck. A DFC attention branch is parallel with the first Ghost module to enhance the expanded features. Then the enhanced features are sent to the second Ghost module for producing output features. It captures the long-range dependence between pixels in different spatial locations and enhances the model's expressiveness.

Table 2: Comparison of SOTA light-weight models over classification accuracy, the number of parameters and FLOPs on ImageNet dataset.

| Model | Params (M) | FLOPs (M) | Top-1 Acc. (%) | Top-5 Acc. (%) |
|---|---|---|---|---|
| MobileNetV1 0.5× [13] | 1.3 | 150 | 63.3 | 84.9 |
| MobileNetV2 0.6× [28] | 2.2 | 141 | 66.7 | - |
| ShuffleNetV1 1.0× (g=3) [42] | 1.9 | 138 | 67.8 | 87.7 |
| ShuffleNetV2 1.0× [22] | 2.3 | 146 | 69.4 | 88.9 |
| MobileNetV3-L 0.75× [12] | 4.0 | 155 | 73.3 | - |
| GhostNetV1 1.0× [8] | 5.2 | 141 | 73.9 | 91.4 |
| GhostNetV1 1.1× [8] | 5.9 | 168 | 74.5 | 92.0 |
| GhostNetV2 1.0× | 6.1 | 167 | **75.3** | **92.4** |
| MobileNetV1 1.0× [13] | 4.2 | 575 | 70.6 | - |
| MobileNetV2 1.0× [28] | 3.5 | 300 | 72.8 | 90.8 |
| ShuffleNetV2 1.5× [22] | 3.5 | 299 | 72.6 | 90.6 |
| FE-Net 1.0× [3] | 3.7 | 301 | 72.9 | - |
| FBNet-B [39] | 4.5 | 295 | 74.1 | - |
| ProxylessNAS [1] | 4.1 | 320 | 74.6 | 92.2 |
| MnasNet-A1 [31] | 3.9 | 312 | 75.2 | 92.5 |
| MnasNet-A2 [31] | 4.8 | 340 | 75.6 | 92.7 |
| MobileNetV3-L 1.0× [12] | 5.4 | 219 | 75.2 | - |
| MobileNeXt 1.0× [44] | 3.4 | 300 | 74.0 | - |
| MobileNeXt+ 1.0× [44] | 3.94 | 330 | 76.1 | - |
| GhostNetV1 1.3× [8] | 7.3 | 226 | 75.7 | 92.7 |
| GhostNetV1 1.4× [8] | 8.2 | 264 | 76.1 | 92.9 |
| GhostNetV2 1.3× | 8.9 | 269 | **76.9** | **93.4** |
| FBNet-C [39] | 5.5 | 375 | 74.9 | - |
| EfficientNet-B0 [32] | 5.3 | 390 | 77.1 | 93.3 |
| MnasNet-A3 [31] | 5.2 | 403 | 76.7 | 93.3 |
| MobileNetV3-L 1.25× [12] | 7.5 | 355 | 76.6 | - |
| MobileNeXt+ 1.1× [44] | 4.28 | 420 | 76.7 | - |
| MobileViT-XS [23] | 2.3 | 700 | 74.8 | - |
| GhostNetV1 1.7× [8] | 11.0 | 378 | 77.2 | 93.4 |
| GhostNetV2 1.6× | 12.3 | 399 | **77.8** | **93.8** |

# 5   Experiments

In this section, we empirically investigate the proposed GhostNetV2 model. We conduct experiments on the image classification task with the large-scale ImageNet dataset [5]. To validate its generalization, we use GhostNetV2 as backbone and embed it into a light-weight object detection scheme YOLOV3 [26]. Models with different backbone are compared on MS COCO dataset [20]. At last, we conduct extensive ablation experiments for better understanding GhostNetV2. The practical latency is measured on Huawei P30 (Kirin 980 CPU) with TFLite tool.

## 5.1   Image Classification on ImageNet

**Setting.** The classification experiments are conducted on the benchmark ImageNet (ILSVRC 2012) dataset, which contains 1.28M training images and 50K validation images from 1000 classes. We follow the training setting in [8] and report results with single crop on ImageNet dataset. All the experiments are conducted with PyTorch [25] and MindSpore [15].

**Results.** The performance comparison of different models on ImageNet is shown in Table 2, Figure 1 and Figure 2. Several light-weight models are selected as the competing methods. GhostNet [8], MobileNetV2 [28], MobileNetV3 [12], and ShuffleNet [42] are widely-used light-weight CNN models with SOTA performance. By combing CNN and Transformer, MobileViT [24] is a new backbone presented recently. Compared with them, GhostNetV2 achieves significantly higher performance with lower computational cost. For example, GhostNetV2 achieves 75.3% top-1 accuracy with only 167 FLOPs, which significantly outperform GhostNet V1 (74.5%) with similar computational cost (167M FLOPs).

Table 3: Results of object detection on MS COCO dataset. YOLOv3 [26] is used as the detection head.

| Backbone | Resolution | Backbone FLOPs (M) | AP | AP$_{50}$ | AP$_{75}$ | AP$_S$ | AP$_M$ | AP$_L$ |
|---|---|---|---|---|---|---|---|---|
| MobileNetV2 1.0× [28] | | 613 | 22.2 | 41.9 | 21.4 | 6.0 | 23.6 | 35.8 |
| GhostNet V1 1.1× | $320 \times 230$ | 338 | 21.8 | 41.2 | 20.8 | 5.7 | 22.3 | 37.3 |
| GhostNetV2 1.0× | | 342 | **22.3** | 41.4 | 21.9 | 6.0 | 22.8 | 38.1 |
| MobileNetV2 1.0× [28] | | 1035 | 23.9 | 45.4 | 22.6 | 10.6 | 25.1 | 34.9 |
| GhostNet V1 1.1× | $416 \times 416$ | 567 | 23.4 | 45.2 | 21.9 | 9.8 | 24.4 | 34.9 |
| GhostNetV2 1.0× | | 571 | **24.1** | 45.7 | 23.0 | 10.4 | 25.0 | 36.1 |

Table 4: Effectiveness of DFC attention with MobileNetV2 on ImageNet dataset.

| Model | Params (M) | FLOPs (M) | Top-1 Acc. (%) | Top-5 Acc. (%) |
|---|---|---|---|---|
| MobileNetV2 1.0 × | 3.5 | 300 | 72.8 | 90.8 |
| MobileNetV2 1.1 × | 4.1 | 338 | 73.0 | 90.0 |
| MobileNetV2 1.1 × + SE [14] | 4.0 | 338 | 73.8 | 91.0 |
| MobileNetV2 1.1 × + CBAM [38] | 4.0 | 338 | 74.0 | 91.4 |
| MobileNetV2 1.1 × + CA [11] | 4.1 | 350 | 74.5 | 91.8 |
| MobileNetV2 1.0 × + DFC (Ours) | 4.3 | 344 | **75.4** | **92.4** |

**Practical Inference Speed.** Considering the light-weight model is designed for mobile applications, we practically measure the inference latency of different models on an arm-based mobile phone, using the TFLite tool [4]. Owing to the deploying efficiency of DFC attention, GhostNetV2 also achieves a good trade-off between accuracy and practical speed. For example, with similar inference latency (*e.g.*, 37 ms), GhostNetV2 achieves 75.3% top-1 accuracy, which is obviously GhostNet V1 with 74.5% top-1 accuracy.

## 5.2 Object Detection on COCO

**Setting.** To validate the generalization of GhostNetV2, we further conduct experiments on the object detection task. The experiments are conducted on MS COCO 2017 dataset, composing of 118k training images and 5k validation images. We embed different backbone into a widely-used detection head, YOLOv3 [26] and follow the default training strategy provided by MMDetection [3]. Specifically, based on the pre-trained weights on ImageNet, the models are fine-tuned with SGD optimizer for 30 epochs. The batchsize is set to 192 and initial learning to 0.003. The experiments are conducted with input resolutions $320 \times 320$.

**Results.** Table 3 compares the proposed GhostNetV2 model with GhostNet V1. With different input resolutions, GhostNetV2 shows obvious superiority to the GhostNet V1. For example, with similar computational cost (*i.e.*, 340M FLOPs with $320 \times 320$ input resolution), GhostNetV2 achieves 22.3% mAP, which suppresses GhostNet V1 by 0.5 mAP. We conclude that capturing the long-range dependence is also vital for downstream tasks, and the proposed DFC attention can effectively endow a large receptive field to the Ghost module, and then construct a more powerful and efficient block.

## 5.3 Semantic Segmentation on ADE20K

We conduct semantic segmentation experiments on ADE20K [43], which contains 20k training, 2k validation, and 3k testing images with 150 semantic categories. We use the DeepLabV3 [2] model as the segmentation head, and follow the default training setting of MMSegmentation [4]. From the pre-trained weights on ImageNet, the models are fine-tuned for 160000 iterations with crop size $512 \times 512$. Table 5 show the results with different backbones. In the semantic tasks, GhostNetV2 also achieves significantly higher performance than GhostNetV1, which illustrates the university of GhostNetV2 over different tasks.

---

[3]https://github.com/open-mmlab/mmdetection.
[4]https://github.com/open-mmlab/mmsegmentation

Table 5: Results of semantic segmentation on ADE20K dataset.

| Backbone | Method | Backbone FLOPs (M) | mIoU (%) |
|---|---|---|---|
| MobileNetV2 1.0× [28] | | 300 | 34.08 |
| GhostNet V1 1.1× | DeepLabV3 | 168 | 34.17 |
| GhostNetV2 1.0× | | 167 | 35.52 |

Table 7: The location for implementing DFC attention.

| Stage | Top1-Acc. (%) | Params (M) | FLOPs (M) |
|---|---|---|---|
| None | 73.9 | 5.2 | 141 |
| 1 | 74.8 | 5.3 | 150 |
| 2 | 75.0 | 5.4 | 152 |
| 3 | 74.7 | 5.8 | 147 |
| All | 75.3 | 5.8 | 168 |

Table 8: Enhancing expressiveness or capacity.

| Model | Top1-Acc. (%) | Params (M) | FLOPs (M) |
|---|---|---|---|
| Baseline | 73.9 (+0.0) | 5.2 | 141 |
| Expressiveness | 75.3 (+1.4) | 6.1 | 167 |
| Capacity | 74.8 (+0.9) | 6.1 | 162 |
| Both | 75.5 (+1.6) | 7.0 | 188 |

## 5.4 Ablation Studies

In this section, we conduct extensive experiments to investigate the impact of each component in GhostNetV2. The experiments are conducted with GhostNetV2 $1\times$ on ImageNet.

**Experiments with other models.** As a universal module, the DFC attention can also be embedded into other architectures for enhancing their performance. The resultsof MobileNetV2 with different attention modules are shown in Table 4. SE [14] and CBAM [38] are two widely-used attention modules, and CA [11] is a SOTA method presented recently. The proposed DFC attention achieves higher performance than these existing methods. For example, the proposed DFC attention improves the top-1 accuracy of MobileNetV2 by 2.4%, which suppresses CA (1.5%) by a large margin.

Table 6: The impact of kernel size in DFC attention.

| Kernel sizes | Top1-Acc. (%) |
|---|---|
| (3, 3, 3) | 74.8 |
| (7, 5, 5) | 75.0 |
| (7, 7, 5) | 74.2 |
| (9, 7, 5) | 75.3 |
| (11, 9, 7) | 75.3 |

**The impact of kernel size in DFC attention.** We split the Ghost-NetV2 architecture into 3 stages by the feature's size, and apply DFC attention with different kernel size (Table 6). The kernel sizes $1 \times 3$ and $3 \times 1$ cannot capture the long-range dependence well, which results in the worst performance (*i.e.*, 74.8%). Increasing the kernel size to capture the longer range information can significantly improve the performance.

**The location for implementing DFC attention.** The GhostNetV2 model can be split into 4 stages by the feature's size, and we empirically investigate how the implementing location affects the final performance. The results are shown in Table 7, which empirically shows that the DFC attention can improve performance when implementing it on any stage. Exhaustively adjusting or searching for proper locations has the potential to further improve the trade-off between accuracy and computational cost, which exceeds the scope of this paper. By default, we deploy the DFC attention on all the layers.

Table 9: The impact of scaling function. 'BF' and 'AF' denote implementing the scaling function before or after the up-sampling operation, respectively.

| Scaling function | Top1-Acc. (%) | FLOPs (M) | Latency (ms) |
|---|---|---|---|
| Sigmoid (BF) | 75.3 | 167 | 37.5 |
| Hard simoid (BF) | 75.2 | 167 | 36.8 |
| Clip (BF) | 74.9 | 167 | 36.7 |
| Sigmoid (AF) | 75.3 | 167 | 40.7 |
| Hard simoid (AF) | 75.2 | 167 | 39.6 |
| Clip (AF) | 75.0 | 167 | 38.5 |

**The impact of scaling function.** For an attention model, it is necessary to scale the feature maps into range (0,1), which can stabilize the training process. Though the theoretical complexity is negligible, these element-wise operations still incur extra latency. Table 9 investigates how the scaling function affects the final performance and latency. Though sigmoid and hard sigmoid functions bring obvious performance improvement, directly implementing them on the large feature maps incur long latency.

Implementing them before up-sampling is much more efficient but results in similar accuracy. By default, we use the sigmoid function and put it before the up-sampling operation.

**Enhancing expressiveness or capacity.** We implement the DFC attention on two Ghost modules and show the results in Table 8. As discussed in Section 4.2, the former enhances expanded features (expressiveness) while the latter improves the block's capacity. With similar computational costs, enhancing the expanded features brings 1.4% top-1 accuracy improvement, which is much higher than enhancing the output feature. Though enhancing both of the features can further improve the performance, the computational cost also increases accordingly. By default, we only enhance the expanded features in an inverse residual bottleneck.

**The resizing functions for up-sampling and down-sampling.** Multiple functions can conduct the up-sampling and down-sampling operations, and we investigate several widely-used functions, *i.e.*, average pooling, max pooling, bilinear interpolation for down-sampling, and bilinear, bicubic interpolations for up-sampling (Table 10). The performance of GhostNetV2 is robust to the choice of resizing functions, *i.e.*, all of these methods achieve similar accuracies in ImageNet. Their differences mainly lie in practical deploying efficiency on mobile devices. Maxing pooling is slightly more efficient than average pooling (37.5 ms *vs.*38.4 ms), and bilinear interpolation is faster than the bicubic one (37.5 ms *vs.*39.9 ms). Thus we choose the maxing pooling for down-sampling and bilinear interpolation for up-sampling by default.

Table 10: The resizing functions for down-sampling and up-sampling, denoted as 'D' and 'U', respectively.

| Resizing function | Top1-Acc. (%) | FLOPs (M) | Latency (ms) |
|---|---|---|---|
| Average Pooling (D) | 75.4 | 167 | 38.4 |
| Max Pooling (D) | 75.3 | 167 | 37.5 |
| Bilinear (D) | 75.3 | 167 | 38.7 |
| Bilinear (U) | 75.3 | 167 | 37.5 |
| Bicubic (U) | 75.4 | 167 | 39.9 |

**Visualization of decoupled attention and full attention.** We visualize the decoupled attention produced by stacking vertical and horizontal attentions and compare it with full attention. In low layers, the decoupled attention shows some cross-shaped patterns, indicating patches from the vertical/horizontal lines participate more. As the depth increases, the pattern of the attention map diffuses and becomes more similar to the full attention.

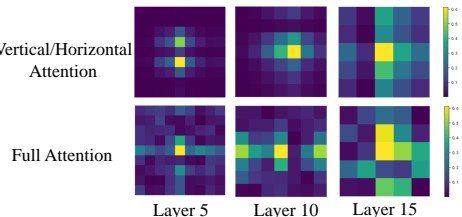

Figure 6: Visualization of attention maps.

## 6 Conclusion

This paper proposes a hardware-friendly DFC attention and presents a new GhostNetV2 architecture for mobile applications. The DFC attention can capture the dependence between pixels in long-range spatial locations, which significantly enhances the expressiveness ability of light-weight models. It decomposes a FC layer into horizontal FC and vertical FC, which has large receptive fields along the two directions, respectively. Equipped this computation-efficient and deployment-simple modules, GhostNetV2 can achieve a better trade-off between accuracy and speed. Extensive experiments on benchmark datasets (*e.g.*, ImageNet, MS COCO) validate the superiority of GhostNetV2.

**Acknowledgment.** This work is supported by National Natural Science Foundation of China under Grant No.61876007, Australian Research Council under Project DP210101859 and the University of Sydney SOAR Prize. We gratefully acknowledge the support of MindSpore, CANN(Compute Architecture for Neural Networks) and Ascend AI Processor used for this research.

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
