# GhostNetV2: Enhance Cheap Operation with Long-Range Attention (Supplementary Material)

**Yehui Tang**[1,2], **Kai Han**[2], **Jianyuan Guo**[2,3], **Chang Xu**[3], **Chao Xu**[1], **Yunhe Wang**[2*]

[1]School of Artificial Intelligence, Peking University [2]Huawei Noah's Ark Lab
[3]School of Computer Science, University of Sydney
yhtang@pku.edu.cn, {kai.han, yunhe.wang}@huawei.com

## 1 More Object Detection Experiments on COCO

**Faster R-CNN.** To validate the generalization of GhostNetV2 with different detection heads, we also embed GhostNetV2 into Faster R-CNN with Feature Pyramid Networks [5, 4]. Following [2], we resize the input images to a short side of 800 and a long side not exceeding 1333. The models are fine-tuned for 12 epochs, and the pre-trained weights on ImageNet are used as the initial values. The results of different backbones are shown in Table A1[2]. Compared with GhostNetV1 (*e.g.*, 26.9 mAP), GhostNetV2 achieves much higher performance (*e.g.*, 28.5 mAP). It empirically shows that GhostNetV2 is compatible with different detection models as a universe backbone.

Table A1: Results of object detection on MS COCO dataset. Faster R-CNN with Feature Pyramid Networks [5, 4] is used as the detection head.

| Backbone | Backbone FLOPs (M) | AP | $AP_{50}$ | $AP_{75}$ | $AP_S$ | $AP_M$ | $AP_L$ |
|---|---|---|---|---|---|---|---|
| MobileNetV2 $1.0\times$ [6] | 300 | 27.5 | 47.4 | 28.5 | 16.0 | 30.6 | 35.2 |
| GhostNet V1 $1.1\times$ | 168 | 26.9 | 46.8 | 27.6 | 16.0 | 29.7 | 34.6 |
| GhostNetV2 $1.0\times$ | 167 | 28.5 | 46.6 | 30.3 | 16.1 | 30.7 | 38.7 |

## 2 Discussion about Decoupled Attention and Full Attention

We compare the calculation process of decoupled attention and full attention in Figure A1. In full attention, all the patches in a region participate in the calculation of the focused patch directly. For the decoupled attention, a patch is directly aggregated by patches in its vertical/horizontal lines, while other patches participate in the generation of those patches in the vertical/horizontal lines, having an indirect relationship with the focused token. Thus the calculation of a patch also involves all the patches in the region. The visualization of attention map is shown in Figure 6 of the main paper.

## 3 More Ablation Studies

**The feature's size in DFC attention.** To reduce the computational cost, we down-sample the feature maps and conduct RA attention on the smaller features. By leveraging the interpolation function, the feature maps can be resized to any shape, whose impact is investigated in Table A2. For a fair comparison, the width of different models is adjusted to control the computational cost. The results show that a better trade-off can be achieved by down-sampling the attention branch, *e.g.*, 75.3% top-1 accuracy with 0.5 down-sampling ratio *vs.*74.4% accuracy without down-sampling. We conclude

---

[*]Corresponding author.
[2]We report the FLOPs of backbone with $224 \times 224$ input images following [2]

36th Conference on Neural Information Processing Systems (NeurIPS 2022).

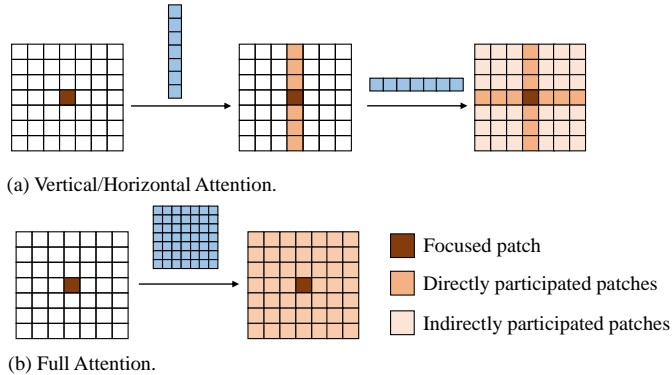

(a) Vertical/Horizontal Attention.

- ■ Focused patch
- ■ Directly participated patches
- □ Indirectly participated patches

(b) Full Attention.

Figure A1: The calculation process of decoupled attention and full attention.

Table A2: The ratio of down-sampling in DFC attention.

| Ratio | Top1-Acc. (%) | Params (M) | FLOPs (G) |
|---|---|---|---|
| 0.3 | 75.2 | 6.8 | 170 |
| 0.5 | 75.3 | 6.1 | 167 |
| 0.7 | 74.9 | 5.5 | 164 |
| 1.0 | 74.4 | 5.0 | 172 |

that the attention branch is to assist the main branch by endowing it long-range dependence, and high-resolution feature maps tend to be redundant in this branch.

**Discussion with NAS-based lightweight non-local networks.** Auto-NL [3] is a NAS-based work following the typical paradigm of self-attention (*i.e.*, $(\boldsymbol{x}\boldsymbol{x}^T)\boldsymbol{x}$, or $\boldsymbol{x}(\boldsymbol{x}\boldsymbol{x}^T)$, where $\boldsymbol{x}$ is a vector), whose computational cost is saved by reducing the feature's dimensions and replacing convolution with light-weight depthwise convolution. It also requires 'einsum', tensor reshaping, and transposing operations for practical implementation, which incur large latency. Since only theoretical FLOPs are reported in [3], we measure its latency using the same devices for GhostNetV2 (Huawei P30 with Kirin 980 CPU) and show the results in Table A3. AutoNL suffers much higher latency (76.4ms v.s. 56.7ms) than GhostNet with lower accuracy (76.5% v.s. 76.9%).

Table A3: Comparison with NAS-based lightweight non-Local networks.

| Model | Top1-Acc. (%) | FLOPs (M) | Latency (ms) |
|---|---|---|---|
| AutoNL-S [3] | 76.5 | 267 | 76.4 |
| GhostNetV2 1.3× | 76.9 | 269 | 56.7 |
| AutoNL-L [3] | 77.7 | 353 | 101.6 |
| GhostNetV2 1.6× | 77.8 | 399 | 77.6 |

NAS-based methods (*e.g.*, Auto-NL [3], OFA [1]) and GhostNetV2 actually focus on different aspects of designing architectures. Auto-NL [3] searches the architecture's configuration (e.g., location for inserting LightNL, channel's number in each layer) to pursue high performance. OFA [1] also searches the architecture configures for specific hardware and uses more training tricks (e.g., progressive shrinking, knowledge distillation) to improve performance. While GhostNetV2 focuses on how to design a hardware-friendly attention mechanism, which doesn't optimize the network architecture and training recipe. Searching network's configuration and improving training recipe have the potential to further improve GhostNetV2's performance.