# OpenReview forum: "GhostNetV2: Enhance Cheap Operation with Long-Range Attention"
_NeurIPS.cc/2022/Conference — NeurIPS 2022 Accept_

### Official Review · Reviewer_L8vJ · 2022-07-04

**Rating:** 7
**Confidence:** 5
**Soundness:** 3 good
**Presentation:** 3 good
**Contribution:** 3 good

**Summary:**

This paper improves a previous work, named GhostNet. The original GhostNet aims to eliminate the effect of those uninformative feature maps by introducing the ghost module. However, the drawback of the original GhostNet is the lack of the ability to capture long-range relationships among pixels. This paper is motivated by this and designs an edge device friendly attention mechanism, which runs fast and performs well on ImageNet classification.

Overall speaking, the quality of this paper is good. The novelty of this paper is significant. Thorough experiments are also conduct to demonstrate the effectiveness of the proposed approach.

**Questions:**

Tables 9 and 10 could be redesigned to make the arrangement look better.

**Limitations:**

Not found.

**Strengths And Weaknesses:**

The strengths  of this paper is clear.

- The proposed light-weight attention mechanism is interesting. It simplifies the standard self-attention and the performance does not drop. I think it could be considered as a promising way to encode global context for mobile networks.

- The analysis of this paper is sufficient. The authors carefully analyze why self-attention is not friendly to edge devices and show lots of results to support this.

- The results on ImageNet are great. Compared to most of previous models for mobile devices, this paper performs better.

Weaknesses:

I do not think there are any major red flags, but some minor to moderate concerns that should be addressed.

- In Sec. 3.1, the authors say that only half of the channels are used for encoding spatial information with a depthwise conv in a ghost module and claim that this may be a performance bottleneck of GhostNet. How could you prove this?

- The performance on downstream tasks, like COCO object detection, seems not that surprising but considering the low computations, it is acceptable.

- The English should be improved, especially the usage of the definite articales.

- The usage of abbreviations. L168 and L183, "Eq 4, 5" should be "Eqs. 4 and 5."

---

> ### Author Response · Authors · 2022-08-02
> **Our response to Reviewer L8vJ**
>
> Thanks for the constructive comments.
>
> **Q1**: In Sec. 3.1, the authors say that only half of the channels are used for encoding spatial information with a depthwise conv in a ghost module and claim that this may be a performance bottleneck of GhostNet. How could you prove this?
>
> **A1**: This is an interesting question. Convolution operations in CNN have a weak ability to capture long-range spatial information. It prevents CNN's performance from further improvement, which is empirically proved by recent works about vision transformers [r1]. As a lightweight CNN, GhostNet even uses smaller kernel sizes (i.e., 1x1 convolution) in half of the channels, which intuitively incurs a weaker ability to model spatial information. This intuition inspires us to design the DFC attention to capture the long-range information, which indeed improves the performance significantly.
>
> **Q2:** The performance on downstream tasks, like COCO object detection, seems not that surprising but considering the low computations, it is acceptable.
>
> **A2:** Thanks for your constructive comments. COCO object detection with lightweight backbones is a challenging task. Compared with the existing architectures, GhostNetV2 achieves higher mAP. In the future, we will continue exploring how to improve the performance of lightweight architectures on downstream tasks.
>
> **Q3**: English and abbreviations.
>
> **A3**: Thanks for your suggestion.  We will polish the writing and fix typos carefully in the final version.
>
> **Q4:** Tables 9 and 10 could be redesigned to make the arrangement look better.
>
> **A4**: Thanks for your suggestion. We will redesign these two tables for better presentation in the final version.
>
> [r1] An Image is Worth 16x16 Words: Transformers for Image Recognition at Scale.

---

> > ### Comment · Reviewer_L8vJ · 2022-08-05
> > **Thanks for the response.**
> >
> > I appreciate the responses from the authors and keep my original rating unchanged regarding the significance of the novelty and the good experimental results.

---

> > > ### Author Response · Authors · 2022-08-07
> > > **Thanks for the comments.**
> > >
> > > Dear Reviewer L8vJ,
> > >
> > > Thanks for your support and constructive comments.
> > >
> > > Regards

---

### Official Review · Reviewer_zuKb · 2022-07-04

**Rating:** 8
**Confidence:** 4
**Soundness:** 4 excellent
**Presentation:** 4 excellent
**Contribution:** 4 excellent

**Summary:**

This paper proposes a hardware friendly attention module and then present a light-weight neural architecture for general vision tasks. It finds that light-weight neural networks have weak ability to capture the global information, which is the bottleneck restricting the representation ability.  The attention module is only constructed by fully-connected layers, which can efficiently capture the global information without complex operations. Then a light-weight architecture is constructed, which achieves SOTA performance on various vision tasks, such as image classification and object detection.

**Questions:**

Please see the weaknesses. Further investigating the impact of non-linear functions will make the paper stronger.

**Limitations:**

Yes

**Strengths And Weaknesses:**

+It is vital to design light-weight neural networks with low latency on edge devices (e.g., ARM CPU) for implementing AI models. Due to the strict constraint on practical latency, improving its performance is a very challenging task. Some complex operations may have low theoretical complexity, but will incur high practical latency as they are not hard-ware friendly. This paper finds the performance bottleneck of light-weight models and presents an efficient DFC attention to capture the global information, which can significantly improve performance.

+The empirical results are impressive. Based on DFC attention, the GhostNetV2 model achieves significantly higher performance (about 1 point) than the existing architectures. Both with attention mechanism, GhostNetV2 achieves much higher performance than MobileViT (77.8% v.s. 74.8%) with lower computational cost (399M v.s. 700M).

+The proposed architecture has strong generalization ability and can be used in diverse tasks (such as image classification and object detection), as it does not introduce prior knowledge of a specific task. Considering its SOTA performance, easy implementation, and strong generalization ability, it can play as the backbone to improve the model’s performance on various tasks.

Though the proposed architecture is novel and effective, I still have some suggestions to further improve its impact on the community.

- The DFC attention captures the long-range dependence with two sequential FC layers, which aggregate information along with horizontal and vertical directions, respectively. Adding a non-linear activation function between them may reflect a more complex relationship between different pixels. It is interesting to investigate the effectiveness of this non-linear function empirically.

-For intuitive presentation, it is better to describe the proposed architectures in the captions of Figure 4.

+This paper explores an interesting direction to leverage the attention mechanism, which may inspire the community. Transformer achieves high performance owing to its strong ability for capturing global information. However, it suffers high computational complexity and complex formulation, which is not hard-ware friendly when implemented on edge devices. The proposed method uses fully-connected layers to implement the attention operation, which is both effective and efficient.

---

> ### Author Response · Authors · 2022-08-02
> **Our response to Reviewer zuKb**
>
> Thanks for the constructive comments.
>
> **Q1**: The DFC attention captures the long-range dependence with two sequential FC layers, which aggregate information along with horizontal and vertical directions, respectively. Adding a non-linear activation function between them may reflect a more complex relationship between different pixels. It is interesting to investigate the effectiveness of this non-linear function empirically.
>
> **A1**: Thanks for your suggestion. We further conduct experiments by inserting a ReLU function between the two FC layers. It slightly improves the performance. For example, the  top-1 accuracy of GhostNetV2 is improved from 75.3% to 75.4%.
>
> **Q2**: For intuitive presentation, it is better to describe the proposed architectures in the captions of Figure 4.
>
> **A2**: Thanks for your suggestion. We will revise the captions to describe the architecture in the final version.

---

> > ### Comment · Reviewer_zuKb · 2022-08-08
> > **After response**
> >
> > Thanks for your response. The rebuttal has well addressed my questions. I support this paper for its novelty and solid experiments. GhostNetV2 is a new and efficient architecture with SOTA performance and great potential. It may inspire new works in the future, such as searching its architecture configures or improving training recipes to pursue higher performance.
> >
> > Thus I vote to accept this manuscript strongly.

---

> > > ### Author Response · Authors · 2022-08-09
> > > **Thanks for the comments.**
> > >
> > > Thanks for your support and constructive comments!
> > >
> > > Regards

---

### Official Review · Reviewer_Fb8Z · 2022-07-11

**Rating:** 6
**Confidence:** 3
**Soundness:** 3 good
**Presentation:** 2 fair
**Contribution:** 3 good

**Summary:**

In this paper, the authors propose a "decoupled fully connected" (DFC) attention module which can significantly improve the performance of image classification models such as GhostNet and MobileNet at a much lower cost than typical self-attention like those used in transformers. This is achieved by aggregating information on the horizontal and vertical axis (instead of over the entire image) in a downsampled feature space and combining the output of this module with the output of standard network block.

Experiments show that this approach empirically achieves significant improvement in accuracy at a modest cost in throughput, and outperforms previous state of the art for efficient mobile-friendly image classification networks.

Nits:

69-70: Until now, GhostNet is still the SOTA light-weight model with a good trade-off between accuracy and speed. -> Until now, GhostNet has been the SOTA light-weight model with a good trade-off between accuracy and speed.

71: Besides manual design, a series of methods try to search a light-weight architecture. -> Besides manual design, a series of methods try to search for a light-weight architecture.

199: practica -> practical

215: Experiment -> experiments

**Questions:**

- This is not quite global attention because you attend over patches aligned horizontally or vertically, and hope that previous blocks captured sufficient global context in their respective patches. one way to test whether they do is to make the network deeper, but narrower (to control for number of params) and see if the accuracy improves. Have you considered this, along with visualizing attention maps to see what is actually being attended

- I am a bit confused about figure 4 and equation 6 and the preceding paragraph. Is it saying that you compute your the horizontal/vertical attention (via a linear projection) followed by sigmoid followed by elementwise multiplication with the output of the ghost module (which is of dim HxWxC). Does this not mean you are just rescaling the output of the ghost module? if not can you please explain and also possibly clarify in the paper how exactly you are attending over/aggregating info from different patches?

**Limitations:**

limitations were addressed adequatly

**Strengths And Weaknesses:**

Strengths:
- Experiments and ablation studies are well designed and show the value of this work
- The results are convincing
- The approach appears to be novel

Weaknesses:
- The description and diagram of the actual approach is somewhat confusing (see the question section)
- It would be good to study what this attention ends up attending over (and compared for vertical/horizontal att vs full att)

---

> ### Author Response · Authors · 2022-08-02
> **Our response to Reviewer Fb8Z**
>
> **Q1**: Nits.
>
> **A1**: Thanks for your careful review. We will fix all the typos in the final version.
>
> **Q2**: The description and diagram of the actual approach are somewhat confusing (see the question section).
>
> **A2**: Thanks for your constructive comments. We detailedly answer the question in **A5**.
>
> **Q3**: It would be good to study what this attention ends up attending over (and compared for vertical/horizontal att vs full att).
>
> **A3**: Thanks for your suggestion. We further visualize the attention of vertical/horizontal attention and full attention in this figure (https://i.postimg.cc/c4tJTKDR/Attention-Visualization.png) and show their calculation process with a diagram (https://i.postimg.cc/13JqKCGd/Diagrams-decoupled-attention-full-attention.png). In full attention, all the patches in a $N \times N$ region participate in the calculation of the focused patch directly. For the decoupled attention, a patch is directly aggregated by patches in its vertical/horizontal lines, while other patches participate in the generation of those patches in the vertical/horizontal lines, having an indirect relationship with the focused token. Thus the calculation of a patch also involves all the patches in the $N \times N$ region.
>
> We visualize the attention produced by stacking vertical and horizontal attentions and compare it with full attention. In low layers, the decoupled attention shows some cross-shaped patterns, indicating patches from the vertical/horizontal lines participate more. As the depth increases, the pattern of the attention map diffuses and becomes more similar to the full attention.
>
> **Q4**: This is not quite global attention because you attend over patches aligned horizontally or vertically, and hope that previous blocks captured sufficient global context in their respective patches. one way to test whether they do is to make the network deeper, but narrower (to control for the number of params) and see if the accuracy improves. Have you considered this, along with visualizing attention maps to see what is actually being attended.
>
>  **A4**: This is an interesting perspective. We make the network deeper and narrower so as to keep FLOPs similar. Their comparisons are shown below, where 'GhostNetV2-x denotes there are x blocks'. Increasing the network's depths can indeed improve the performance, and we infer that the long-range information along the two directions can be mixed more thoroughly as the depth increase. Over increasing depth does harm to the performance as the long-range information is saturated but the channels are too few. This phenomenon is consistent with the attention's visualization and analysis in **A3**.
>
> | Model | GhostNetV2-16 |   GhostNetV2-20   | GhostNetV2-25 |   GhostNetV2-30   |
> | ------------------ | ------------------------------ | ---- | ---- | ---- |
> | Accuracy | 75.3 | 75.7 | 75.6 | 75.1 |
>
> **Q5**: I am a bit confused about figure 4 and equation 6 and the preceding paragraph. Is it saying that you compute your horizontal/vertical attention (via a linear projection) followed by sigmoid followed by elementwise multiplication with the output of the ghost module (which is of dim HxWxC). Does this not mean you are just rescaling the output of the ghost module? if not can you please explain and also possibly clarify in the paper how exactly you are attending over/aggregating info from different patches?
>
> **A5**: Thanks for your suggestion. We further clarify the information aggregation process by this figure (https://i.postimg.cc/Vkjtbsvz/Fusion.png). With the same input, the Ghost module and DFC attention are two parallel branches extracting information from different perspectives.  The output is their element-wise product, which contains information from both features of the Ghost module and attentions of the DFC attention module. The calculation of each attention value involves $N \times N$ patches in a large range so that the output feature can contain information from these patches.

---

> > ### Comment · Reviewer_Fb8Z · 2022-08-08
> > **Response**
> >
> > Thank you for your clarifying comments. I believe this paper would be interesting to the community, especially if the paper is later updated to include the additional explanations and make the writing / diagrams a bit more clear. I have increased my score as a result.

---

> > > ### Author Response · Authors · 2022-08-09
> > > **Thanks for your feedback!**
> > >
> > > Dear Reviewer Fb8Z:
> > >
> > > Thanks for your feedback and valuable suggestions! We have revised the manuscript and supplemental materials. We improve our writing and include additional explanations, diagrams, and discussions to make the paper clear.
> > >
> > > Regards

---

### Official Review · Reviewer_BfQb · 2022-07-14

**Rating:** 3
**Confidence:** 3
**Soundness:** 2 fair
**Presentation:** 3 good
**Contribution:** 2 fair

**Summary:**

This paper proposes a cheap attention module that focuses on mobile settings, e.g. limited FLOPs and latency. The cheap attention module is implemented as two depthwise convolutions, followed by sigmoid attention. The module is mainly used to augment GhostNet and achieves improvements on ImageNet classification and downstream tasks, e.g. detection, and segmentation.

**Questions:**

Minor: should the latter $a$ be $a\prime$ in equation 5?

Also, in equation 4, for each h, the equation does not depend on h. Does it mean that the output $a\prime$ is the same for each h? If so, equation 5 does not depend on w?

**Limitations:**

Yes.

**Strengths And Weaknesses:**

### Strengths
1. There are many ablation studies that validate the model design choices.
2. Extensive experiments on multiple large scale tasks and datasets.

### Weaknesses
1. The proposed solution, Equation 4,5, is not depthwise convolution with kernel $K_H$ and $K_W$. In equation 4 and 5, the weight matrix F has HWC parameters. These parameters are not shared for each column or row. This is more like a batch matrix multiplication, but is different from depthwise convolution which has $K_H$ and $K_W$ parameters. According to table 5, it seems that depthwise convolution is being used in practice.
2. Limited novelty with height-width decoupled depthwise convolutions [spatially separable convolution] and SE-like spatial-channel attention. Both of the techniques have been heavily explored in the community.
3. Marginal performance gain when FLOPs is controlled. For example, Auto-NL [1] also studied lightweight self-attention, achieved 77.7% on ImageNet with 353M FLOPS and 5.6M parameters two years ago, compared with this paper, 77.8% with 399M FLOPs and 12.3M parameters. Although it is not clear what latency Auto-NL needs, this related work is not discussed or compared.
4. Downsampling and upsampling part of the feature map has been explored too in [2]. And it leads to gains as well. So it is not clear why DFC works.
5. Latency is important to the paper, so instead of saying “an ARM-based mobile device”, is it possible to specify which device exactly, then it is possible for others to reproduce the latency results and make meaningful comparisons.
6. Related to the point above, FBNet reports a latency of 28.1ms, but this paper reports around 70ms. Is this mainly caused by device difference and optimization? Also, [3] reports multiple models on multiple mobile devices, e.g. 76.1% with 22ms on Samsung Note 10, or 76.9% with 58ms on Pixel 1. Is it possible to compare the latency results with the literature? Similarly, MobileViT reports a 17.86ms latency on iPhone 12, which is much faster than the GhostNet+self attention baseline in Table 1, probably due to a larger feature resolution in the comparison.

[1] Neural Architecture Search for Lightweight Non-Local Networks, CVPR 2020.
[2] ELASTIC: Improving CNNs with Dynamic Scaling Policies, CVPR 2019.
[3] Once for All: Train One Network and Specialize it for Efficient Deployment, ICLR 2020.

---

> ### Author Response · Authors · 2022-08-02
> **Our response to Reviewer BfQb (Part 2/2)**
>
> **Q4**: Downsampling and upsampling part of the feature map has been explored too in [r2]. And it leads to gains as well. So it is not clear why DFC works.
>
> **A4**: Feature downsampling is actually a direct idea when the computational cost is excessive, and here we adopt it to reduce DFC attention's computational cost. In [r2], half of the features are down-sampled in block level to reduce FLOPs, while our DFC attention downsamples the features in the attention path. We apply Elastic method on GhostNetV1 to verify its effectiveness on mobile networks. The results are shown below, where the model's width is adjusted to align the FLOPs.  With similar FLOPs, downsampling a part of features (GhostNetV1 + Elastic) can improve the performance, but it is much inferior to that of using DFC attention. We infer that DFC attention can capture long-range information and improve performance more effectively.
>
> | Model               | FLOPs (M) | Top-1 Accuracy (%) | Top-5 Accuracy (%) |
> | ------------------- | --------- | ------------------ | ------------------ |
> | GhostNetV1          | 168       | 74.5               | 92.0               |
> | GhostNetV1+ Elastic | 172       | 74.8               | 92.1               |
> | GhostNetV2          | 167       | 75.3               | 92.4               |
>
> **Q5**: Latency is important to the paper, so instead of saying “an ARM-based mobile device”, is it possible to specify which device exactly, then it is possible for others to reproduce the latency results and make meaningful comparisons.
>
> **A5**: Thanks for your suggestion. The practical latency is measured on Huawei P30 (Kirin 980 CPU) with TFLite tool.
>
> **Q6-1**: Related to the point above, FBNet reports a latency of 28.1ms, but this paper reports around 70ms. Is this mainly caused by device difference and optimization?
>
> **A6-1**: Yes, the devices and implementation tools usually incur a large difference in practical latency, e.g., OFA [r3] (Figure 10) reports 76.3% top-1 with 89ms on Samsung S7 Edge, and 76.4% top-1 with 58ms on Pixel 1. It is hard to compare the model's latency with different devices and implement tools. Moreover, the available devices are different for various companies or institutions, e.g., Google usually implements models on Pixel series phones [r4], while Apple uses iPhones [r5]. Thus a widely-used comparison strategy is to measure different models' speeds on the same device as we did in the paper.
>
> **Q6-2**: Also, [r3] reports multiple models on multiple mobile devices, e.g. 76.1% with 22ms on Samsung Note 10, or 76.9% with 58ms on Pixel 1. Is it possible to compare the latency results with the literature?
>
> **A6-2**: OFA [r3] is a latency-aware NAS method that searches the architecture configures for specific hardware. It also uses more training tricks (e.g., progressive shrinking, knowledge distillation) to improve performance. While our method is to propose a universe module without optimizing for a specific device. These two methods focus on different aspects and have the potential to be combined.
>
> **Q6-3**: Similarly, MobileViT reports a 17.86ms latency on iPhone 12, which is much faster than the GhostNet+self attention baseline in Table 1, probably due to a larger feature resolution in the comparison.
>
> **A6-3**: For both “GhostNet+Self attention” and “GhostNet+ DFC Attention”, we both use the standard input’s resolution of ImageNet, i.e., 224x224. Thus their comparison is fair. MobieViT [r5] implements the model with CoreML on iPhone 12, where  MobileViT-XS is much slower than MobileNetV2 (Table 11 in [r5]). MobileViT-XS achieves 74.8% accuracy with 700M FLOPs, while our GhostNetV2 1.3$\times$ achieves higher performance (76.9%) with a much lower computational cost (269M).
>
> **Q7 (minor)**: Should the latter $a$ be $a'$ in equation 5?
>
> **A7**: Thanks for your careful review. The latter $a$ should be $a'$ in Eq. 5 and we will fix it in the final version.
>
> **Q8 (minor)**: In equation 4, for each h, the equation does not depend on h. Does it mean that the output is the same for each h? If so, equation 5 does not depend on w?
>
> **A8**: The transformation weights in Eqs. 4 and 5 are $F^H_{h,h'w}$ and $F^W_{w,hw'}$, respectively. So Eq. 4 depends on h and Eq. 5 depends on w. Sorry for the typo.
>
> [r1] Neural Architecture Search for Lightweight Non-Local Networks, CVPR 2020.
>
> [r2] ELASTIC: Improving CNNs with Dynamic Scaling Policies, CVPR 2019.
>
> [r3] Once for All: Train One Network and Specialize it for Efficient Deployment, ICLR 2020.
>
> [r4] MobileNetV2: Inverted Residuals and Linear Bottlenecks, CVPR 2018.
>
> [r5] MobileViT: Light-weight, General-purpose, and Mobile-friendly Vision Transformer, ICLR 2022.

---

> ### Author Response · Authors · 2022-08-02
> **Our response to Reviewer BfQb (Part 1/2)**
>
> **Q1**: The proposed solution, Equation 4,5, is not depthwise convolution with kernel $K_H$ and $K_W$. In equation 4 and 5, the weight matrix F has HWC parameters. These parameters are not shared for each column or row. This is more like a batch matrix multiplication, but is different from depthwise convolution which has $K_H$ and $K_W$ parameters. According to table 5, it seems that depthwise convolution is being used in practice.
>
> **A1:** Eqs. 4 and 5 denote the general formulation of DFC attention, which aggregates pixels along horizontal and vertical directions, respectively. The implementation strategy is discussed in Line 168-174 of the submitted manuscript. With weight sharing, depth-wise convolution with kernel $1\times K_H$ and $K_W\times1$ can accomplish the aggregating process along the two directions as Eqs. 4 and 5. This strategy is well supported by tools such as TFLite and ONNX for fast inference on mobile devices.
>
> **Q2**: Limited novelty with height-width decoupled depthwise convolutions [spatially separable convolution] and SE-like spatial-channel attention. Both of the techniques have been heavily explored in the community.
>
> **A2**: This paper discusses a practical and important problem, i.e., how to design a spatial attention mechanism for efficient architectures. It is required to capture long-range spatial dependency and be efficiently deployed on mobile devices as well, while the existing methods cannot satisfy them simultaneously. Though the proposed DFC attention is concept-simple, it satisfies the two properties and helps develop a new GhostNetV2 architecture with higher performance and lower latency. We argue that the discussion about designing hardware-friendly spatial attention can bring new perspectives to the community and the proposed GhostNetV2 architecture can be practically applied to various mobile devices.
>
> The representative SE-like attentions include SE[11], CBAM [31] and CA [8]. There are many differences between the proposed DFC attention and these methods. For example, instead of global pooling along height or width in SE-like attentions, our DFC attention keeps the H/2xW/2 size which is beneficial for spatial and fine-grained information. Besides, the horizontal and vertical attentions are conducted sequentially, which can involve patches more efficiently than the conventional parallel formulation.  Our DFC attention is compared with these SE-like attentions in Table 4 of the submitted manuscript, which outperforms SE-like attentions by a significant margin.
>
> **Q3**: Marginal performance gain when FLOPs is controlled. For example, Auto-NL [r1] also studied lightweight self-attention, achieved 77.7% on ImageNet with 353M FLOPS and 5.6M parameters two years ago, compared with this paper, 77.8% with 399M FLOPs and 12.3M parameters. Although it is not clear what latency Auto-NL needs, this related work is not discussed or compared.
>
> **A3**:  Thanks for your suggestion.  Auto-NL [r1] is a nice work, and we further compare our method with it. Auto-NL follows the typical paradigm of self-attention (i.e., $(xx^T)x$ or $x(x^Tx)$), whose computational cost is saved by reducing the feature's dimensions and replacing $1\times1$ convolution with light-weight depthwise convolution. It also requires 'einsum', tensor reshaping, and transposing operations for practical implementation, which incur large latency. Since the original paper [1] only reports theoretical FLOPs without practical latency, we measure its latency using the same devices for GhostNetV2 (Huawei P30 with Kirin 980 CPU) and show the results as follows. AutoNL suffers much higher latency (76.4ms v.s. 56.7ms) than GhostNet with lower accuracy (76.5% v.s. 76.9%).
>
> | Model           | FLOPs (M) | Latency (ms) | Top-1 Accuracy (%) | Top-5 Accuracy (%) |
> | --------------- | --------- | ------------ | ------------------ | ------------------ |
> | AutoNL-S        | 267       | 76.4         | 76.5               | 93.1               |
> | GhostNetV2 1.3× | 269       | 56.7         | 76.9               | 93.4               |
> | AutoNL-L        | 353       | 101.6        | 77.7               | 93.7               |
> | GhostNetV2 1.6× | 399       | 77.6         | 77.8               | 93.8               |
>
> Besides, Auto-NL and GhostNetV2 actually focus on different aspects of designing architectures. Auto-NL is a NAS-based method, which searches the architecture's configuration (e.g., location for inserting LightNL, channel's number in each layer) to pursue high performance. While GhostNetV2 focuses on how to design a hardware-friendly attention mechanism, which doesn’t optimize the network architecture.  Searching network's configuration has the potential to further improve the performance of GhostNetV2, but it may be out of this paper's scope.

---

> ### Author Response · Authors · 2022-08-07
> **Thanks for the comments.**
>
> Dear Reviewer BfQb,
>
> Thanks for your constructive review. Has our response resolved your concerns? If there are other questions, we are glad to discuss them with you.
>
> Regards

---

### Author Response · Authors · 2022-08-09
**Updated revision of the manuscript**

Dear area chair and anonymous reviewers,

Thanks for your constructive comments and valuable suggestions to improve this paper. We have revised the manuscript and supplemental materials by improving the presentation and including more experiments, discussions, and explanations. If you have any questions, we are glad to discuss them with you.

Regards

---

### Meta-Review · Area_Chair_Qu1L · 2022-08-26

**Recommendation:** Accept
**Confidence:** Certain

**Metareview:**

This paper aims to augment efficient CNNs with self-attention. However, since the naive approach to self-attention is computationally expensive and would contradict the point of efficient CNNs, the authors introduce a new attention mechanism which captures long-range information without substantially added computation cost. The paper demonstrates that GhostNetV2 exhibits markedly better performance at various compute limits as compared to previously proposed efficient networks. Three of the reviewers were quite positive on this paper, noting the novelty of the approach and the strength of the empirical results. One reviewer had several concerns, primarily regarding comparison to NAS based approaches and the novelty of the approach. I agree with the other reviewers that it is not reasonable to compare NAS approaches to non-NAS approaches, and agree that there are marked differences between this work and the previous work cited. I therefore recommend acceptance. I think this will be a valuable contribution to the efficient network community.

**Award:**

No

---

### Decision · Program_Chairs · 2022-09-14

Accept